Asexual reproduction of a few genotypes favored the invasion of the cereal aphid Rhopalosiphum padi in Chile

Rubio-Meléndez María E. 1 2 3
Barrios-SanMartin Joceline 1 2
Pina-Castro Felipe E. 1 2
http://orcid.org/0000-0001-9218-5564 Figueroa Christian C. 1 2
Ramirez Claudio C. 1 2 clramirez@utalca.cl
1 Centre for Molecular and Functional Ecology in Agroecosystems, Universidad de Talca , Talca, Maule , Chile
2 Instituto de Ciencias Biológicas, Universidad de Talca , Talca, Maule , Chile
3 Centro de Bioinformática y Simulación Molecular, Facultad de Ingeniería, Universidad de Talca , Talca, Maule , Chile
Gillespie Joseph
Electronic publication date: 2019 Jul 26
Publication date: 2019
Volume: 7
Electronic Location ID: e7366
Received 2019 Mar 25; Accepted 2019 Jun 27
Copyright: © 2019 Rubio-Meléndez et al.
Copyright year: 2019
Copyright holder: Rubio-Meléndez et al.
License: This is an open access article distributed under the terms of the Creative Commons Attribution License, which permits unrestricted use, distribution, reproduction and adaptation in any medium and for any purpose provided that it is properly attributed. For attribution, the original author(s), title, publication source (PeerJ) and either DOI or URL of the article must be cited.
License URL: https://creativecommons.org/licenses/by/4.0/

Keywords: Multilocus genotypes, Superclone, Genetic diversity, Population, Pest aphids, Genetic diversity, Cereals, Biological invasion

Funding: Fondo Nacional de Desarrollo Científico y Tecnológico (Fondecyt Regular) 1131008 Chilean Ministry of Economy through Iniciativa Científica Milenio NC120027 This work was funded by the Fondo Nacional de Desarrollo Científico y Tecnológico (Fondecyt Regular Grant No. 1131008 to Claudio C. Ramirez and Fondecyt Postdoc Grant No. 3190544 to María Rubio-Meléndez) and the Chilean Ministry of Economy through Iniciativa Científica Milenio (Grant NC120027). There was no additional external funding received for this study. The funders had no role in study design, data collection and analysis, decision to publish, or preparation of the manuscript.

==============================
Background

Aphids (Hemiptera: Aphididae) are insects with one of the highest potentials for invasion. Several aphid species are present globally due to introduction events; they represent important pests of agroecosystems. The bird cherry-oat aphid Rhopalosiphum padi (Linnaeus) is a major pest of cereals and pasture grasses worldwide. Here, we report the genetic features of populations of R. padi that colonize different cereal crops in central Chile.

Methods

Rhopalosiphum padi individuals were collected in central Chile and genotyped at six microsatellite loci. The most frequent multilocus genotype (MLG) was then studied further to assess its reproductive performance across cereal hosts under laboratory conditions.

Results

Populations of R. padi in Chile are characterized by a low clonal diversity (G/N = 62/377 = 0.16) and the overrepresentation of a few widely distributed MLGs. One of the MLGs constituted roughly half of the sample and was observed in all sampled populations at high frequencies. Furthermore, this putative aphid “superclone” exhibited variations in its reproductive performance on cereals most commonly cultivated in Chile. The sampled populations also exhibited weak signs of genetic differentiation among hosts and localities. Our findings suggest that (1) obligate parthenogenesis is the primary reproductive mode of R. padi in Chile in the sampled range and (2) its introduction involved the arrival of a few genotypes that multiplied asexually.

Introduction

Worldwide, roughly 30% of crops are lost due to pests and pathogens. Worryingly, these yield losses are expected to increase due to global warming, particularly for major grain crops (e.g., wheat, rice, and maize) in temperate regions (Riegler, 2018). Insect pests are projected to cause an additional 10–25% of grain yield losses for each degree of global mean surface warming (Deutsch et al., 2018). This trend arises because insect pests feed more in warmer climates and rapidly multiply. As a result, biological control is less effective, and invasion events are more likely (Colinet et al., 2015; Selvaraj & Ganeshamoorthi, 2013; Bebber, 2015).

Aphids (Hemiptera: Aphididae) are insects with one of the highest potentials for invasion. Several pest aphids are present globally due to introduction events, and they accordingly represent important pests in agroecosystems worldwide. The biological features that explain the aggressiveness and invasiveness of aphids include: (i) parthenogenetic reproduction that enables rapid growth of the population, (ii) the global availability of cultivated and wild hosts, and (iii) their ample phenotypic plasticity in response to changing environments (Figueroa et al., 2018).

The bird cherry-oat aphid Rhopalosiphum padi (Linnaeus) is a major pest of cereal crops worldwide (Van Emden & Harrington, 2017). Given that R. padi aphids mostly reproduce via female-only asexual reproduction during the spring, asexual lineages can rapidly multiply and spread by alate forms colonizing different host plants. This situation results in what has been called “genetic inflation” (Loxdale et al., 2017). This aphid transmits the most damaging strains of the barley yellow dwarf virus, resulting in significant losses to wheat yields (up to 40% in some primary wheat-production areas) (Girvin et al., 2017). It is projected that densities of R. padi will increase in warmer climates, which favor aphid dispersal and virus transmission (Ryalls & Harrington, 2016; Claflin, Power & Thaler, 2017). In addition, R. padi has evolved both metabolic and target‐site mechanisms of resistance to several synthetic insecticides used for its control (e.g., organophosphates, carbamates, and neonicotinoids), particularly in important areas of wheat production in Asia where this aphid has rapidly become the most frequent cereal pest (Chen et al., 2007; Wang et al., 2018a, 2018c). Therefore, R. padi represents a serious present and future threat to food security.

Depending on the availability of their primary host, three main types of life-cycles can be displayed by R. padi (Simon, Blackman & Le Gallic, 1991): (1) cyclic parthenogenesis, also referred to as holocyclic, with several parthenogenetic generations on Poaceae (e.g., cereals and pastures) alternated by a single sexual generation on Prunus padus and P. virginiana during the autumn, (2) obligate parthenogenesis, with only asexual lineages produced all year on Poacea, and (3) androcyclic, with several parthenogenetic generations on Poacea followed by the production of only sexual males in the autumn. As described for several pest aphids, the predominance of a certain reproductive mode in R. padi relies on geographic and climatic factors (Halkett et al., 2004; Gilabert et al., 2009; Rispe et al., 1998). For example, a positive correlation between the number of sexual individuals produced by R. padi and winter severity has been found (Halkett et al., 2004). While cyclic parthenogenesis is the preferred reproduction mode for aphid populations living in regions with cold winters, populations of pest aphids in temperate regions are mostly composed of obligate parthenogenesis lineages (Simon & Peccoud, 2018).

Understanding the genetic features of pest populations can help inform decisions about appropriate pest-management strategies. By using high-resolution molecular markers (e.g., microsatellite loci), it is possible to determine basic aspects of pest biology and anticipate their evolutionary responses (Simon & Peccoud, 2018). Species and clone identifications of aphids are provided by these markers, as well as the genetic structure of pest populations, which inform about the strength of forces driving the microevolutionary process of aphid populations such as founder effects, bottlenecks and migratory event (Loxdale et al., 2017). One special feature of most introduced pest aphids worldwide is that their populations are dominated by a few obligate parthenogenetic genotypes and characterized by low genetic and genotypic diversities (Figueroa et al., 2018). Agroecosystems are quite homogeneous in space and time due to similar agricultural practices, the availability of cultivated and wild hosts, and the cultivation of certain crops under similar climates, among others (Van Emden & Williams, 1974). Therefore, introduced aphids are not faced with strong environmental variations, which may explain why obligate parthenogenetic genotypes that invade similar agroecosystems worldwide (e.g., wheat fields) may exhibit similar phenotypes and rapidly multiply until they become superclones.

Here, we studied genetic diversity in populations of R. padi in the primary cereal-growing region of Chile. By sampling R. padi individuals from main cereals and genotyping with six microsatellite loci, we showed that these populations are characterized by low genetic diversity, strong indications of the presence of obligate parthenogenetic as a unique reproductive mode, and the dominance of one single genotype with features of a superclone. We also investigated this single genotype in the laboratory to determine whether this genotype, collected from different locations and host plants, exhibited phenotypic plasticity in reproduction across different cereal hosts. Our results may facilitate effective implementation of pest-management strategies of invasive pests, particularly those with predominant clonal reproduction.

Materials and Methods

Aphid collection and laboratory rearing

Rhopalosiphum padi individuals were collected from 17 different cereal fields located in central Chile (Table 1). Sampling were carried out during early spring-early summer 2013–2014, early spring-early summer 2014–2015, and early spring 2015. Mid and late summer samples were no taken to avoid the effect of clonal selection or drift possibly causing shifts in the relative frequencies of clones throughout the season (Vorburger, 2006). The sampling was conducted through a latitudinal transect of ca 230 km from 33°S to 38°S, an area characterized by a dry Mediterranean climate. During the first sampling campaign (early spring of 2013) randomly selected private fields containing cereal crops, where sampling was allowed, were monitored for R. padi. During the following years, the same sets of fields were revisited and sampled in case of finding the same crop. Sampled hosts included those most commonly cultivated in the area: wheat (Triticum turgidum L. subsp. durum), barley (Hordeum vulgare), oat (Avena sativa), and maize (Zea mays). Aphids were collected from colonies growing on a single host plant (one single leaf per plant). Samples were separated by at least 10 m from the next sampled plant, thus limiting the chance of taking individuals from the same parthenogenetic colony. Samples were gently taken with a paintbrush and stored in tubes of 1.7 mL filled with absolute ethanol. They all were identified under a binocular microscope following taxonomic keys in the laboratory (Blackman & Eastop, 2000; Dixon & Glen, 1971). This was necessary in order to exclude the corn aphid R. maidis in further analyses. Thus, a total of 377 colonies of R. padi were sampled (Table 1). Only one apterous individual per colony was used for further determination and DNA isolation. In addition, live individuals were collected on each sample site were established in the laboratory as clonal lines for aphid performance assessment (see below). Therefore, between one to 10 individuals per sampled colony were taken to the laboratory and reared on barley (H. vulgare cv. Sebastián). All aphid colonies were reared in a system composed by potted plants enclosed in a plastic-transparent cylinder covered with mesh cloth fabric on top that facilitate air circulation and avoid aphid from escaping. Each lineage was maintained in triplicated in these plastic cylinders inside closed acrylic cages (50 × 40 × 30 cm). Aphids were then left to multiply by parthenogenesis, as they were reared under conditions that ensure their asexual reproduction. These units were maintained under controlled environmental conditions in a 4.0 × 5.0 m growth chambers (20 ± 2 °C, 80% humidity, 36.8 μmol/m2/s PAR as measured with LI-190R Quantum Sensor and 16 light:8 dark photoperiod). Standard fluorescent tubes combined with 600W high pressure sodium lamp were used to ensure constant radiation. For the study of the performance of the most common genotype (see below), clonal lineages of these genotype were maintained separately in the laboratory according to the samples site, generating different clonal lineages of the same genotype, which were compared in terms of the reproductive performed across different cereal hosts.

Table 1 Information of Rhopalosiphum padi sampling on cereal crops in central Chile.

The number of correctly determined aphids per sampling site and growing season is shown. These samples were further used for microsatellite genotyping. Dates with effective sampling are recorded.

Sampling locality	Acronym	GPS coordinates (Lat./Long.)	Sampling date	Host plant	Number of colonies	
María Pinto	MP	33°30′07″/71°07′34″	November 2013	T. turgidum	30	
Melipilla	ME	33°56′09″/71°24′32″	November 2014	T. turgidum	21	
San Vicente	SV	34°02′53″/71°22′48″	November 2014	T. turgidum	25	
Lago Rapel	LR	34°14′09″/71°23′36″	November 2014	T. turgidum	25	
Las Cabras	LC	34°13′49″/71°18′08″	November 2014	T. turgidum	21	
Santa Cruz	SCR	34°38′18″/71°22′57″	October 2013	T. turgidum	30	
Rauco	RA	34°57′01″/71 20′27″	January 2014	Z. mays	12	
Licantén	LI	35°01′58″/72°07′12″	November 2013/September 2014	T. turgidum	53	
Limavida	LIM	35°01′01′/71°46′60″	August 2015	T. turgidum	4	
Docamavida	DO	35°00′00′/71°57′00″	August 2015	H. vulgare	4	
Curepto	CU	35°02′32″/72°04′15″	October 2013	T. turgidum	10	
Villa Prat	VP	35°06′17″/71°37′02″	October 2013/September 2014	T. turgidum	67	
Pelarco	PE	35°22′48″/71°34′08″	October 2013/November 2014	T. turgidum	46	
San Clemente	SCL	35°33′38″/71°27′26″	September 2013	T. turgidum	16	
Queri	QE	35°35′52″/71°24′47″	August 2013	A. sativa	7	
Villa Alegre	VA	35°44′15″/71°42′31″	August 2015	T. turgidum	3	
Cato	CA	37°38′75″/72°35′30″	August 2014	T. turgidum	3	

Microsatellite genotyping

Genomic DNA was extracted from each individual aphid using the “salting-out” method (Sunnucks & Hales, 1996). Extracted DNA was quantified using NanoDrop spectrophotometer (EEUU; Thermo Scientific, Waltham, MA, USA). Species-specific microsatellite loci isolated by Simon et al. (2001) were selected for genotyping because they were reported to amplify successfully in previous studies (Delmotte et al., 2002; Wilson et al., 2004; Valenzuela et al., 2010; Gilabert et al., 2009; Duan et al., 2017). Loci R1.35, R5.10, R2.73, R5.29b, R6.3, R3.171, R5.138, and R5.50 were tested (see primer in Table S1). Primers for R1.35 and R6.3 loci failed to show amplification. Afterward, a fluorescent labeling polymerase chain reaction (PCR) protocol was set-up according to Schuelke (2000). PCR reactions were carried out in 15 μL reaction volume containing 0.3 μL of forward primer, 0.9 μL of reverse primer, and 0.9 μL of M13 primer labeled with a FAM or VIC fluorescent dye, 100 μM of dNTPs, 1× Mg++ free reaction buffer, 50 mM of MgCl2, 0.5 U of Platinum Taq DNA Polymerase (Invitrogen, Carlsbad, CA, USA) five U/μL, and three to six ng/μL of DNA. Dyes were assigned to each locus in a way that allowed us to amplify all six loci in only three multiplex PCRs (multiplex 1: R5.50 (FAM) and R5.138 (VIC); multiplex 2: R5.10 (FAM) and R5.29b (VIC); multiplex 3: R3.171 (FAM) and R2.73 (VIC). The thermal cycling conditions consisted of 2 min of initial denaturation at 94 °C followed by 32 cycles of 20 s at 94 °C, 20 s at 58 °C, and 20 s at 72 °C, with additional eight cycles of 30 s at 94 °C, 30 s at 53 °C, and 45 s at 72 °C, and a final elongation step of 2 min at 72 °C (Simon et al., 2001). All PCR reactions were run in a Viriti Thermal Cycler (Applied Biosystems, Foster City, CA, USA). The size of the amplicons was analyzed in Macrogen Inc. (Seoul, Korea) by the capillary sequencer (ABI 3130xl; Applied Biosystems, Foster City, CA, USA). To avoid cross-contamination among samples during shipping, 96-well plates containing PCR products were sealed with microtube caps and negative control with no DNA were also included. Electropherograms were analyzed using the GeneMaker software (Softgenetics, State College, PA, USA) (Hulce, Li & Snyder-Leiby, 2011), with GeneScan-500 LIZ internal standards for allele sizing. Null alleles (An) were checked using MICRO-CHECKER version 2.2.3 software (Van Oosterhout et al., 2004).

Multilocus genotypes

The multilocus genotype (MLG) for each individual aphid was obtained by combining the alleles amplified from all six microsatellite loci. We assume that individuals carrying the same combination of alleles (i.e., the same MLG) have inherited its genetic architecture from a genetically identical asexual ancestor (i.e., clones); eventually, however, random mutations may produce changes in the length of short sequence repeats at a certain locus (i.e., changes in the size of microsatellite alleles) within asexual lineages (Loxdale, 2008). This means that those MLGs sharing most of their alleles and which are also similar in size, have a better chance to have evolved from the same ancestral lineage than from another unrelated genotype. Those assumptions need to be considered, as aphids were reproducing exclusively by parthenogenesis at the time they were sampled (i.e., Austral spring). This allowed us to characterize and compare the clonal diversity and genotypic composition of R. padi within and between sampling sites. The probability that replicates of the same MLG are products of different sexual reproductive events was calculated using PSEX statistic as implemented in MLGsim 2.0, an updated version of MLGsim (Stenberg, Lundmark & Saura, 2003) facilitated by Dr. Aniek Ivens.

Genetic diversity

The genetic diversity of R. padi was computed using different indices. To assess the magnitude of the distortions that the over representation of some aphid clones can produce on measuring the genetic diversity, data analyses were performed on both the whole sample (i.e., including all the clonal copies) and on one single copy per MLG, and then compared among populations. Thus, we focused on the frequency of each MLG rather than the frequency of microsatellite alleles (Figueroa et al., 2005). The gross genotypic diversity was estimated as Pd = G/N, where G is the number of MLGs found and N the total number of individuals genotyped. The clonal heterogeneity indices (D*, adapted Simpson’s index and Hill’s Simpson’s reciprocal index) and clonal evenness index (ED*, Simpson’s evenness index) were computed using the software GENCLONE 2.0 (Arnaud‐Haond & Belkhir, 2007). In order to assess the expected frequencies for each MLG in every population, the Hardy–Weinberg equilibrium was calculated according to Brookfield (1996) using GENEPOP package version 1.2 (Raymond & Rousset, 1995; see http://genepop.curtin.edu.au/). The linkage disequilibrium (LD) between pairs of loci and the inbreeding coefficient (Fis) over all loci were computed in the same software (log likelihood ratio statistic with 10,000 dememorization number, 10,000 iterations per batch, and 100 batches), using only one copy per MLG (Halkett et al., 2005). In addition, we estimated the proportion of An, the mean number of alleles per locus (n) and the mean expected (HE) and observed (HO) heterozygosity over all loci for each population using GENALEX 6 (Peakall & Smouse, 2006; see http://biology-assets.anu.edu.au/GenAlEx/Welcome.html). The allelic richness was calculated using FSTAT version 2.9.3.2 (Goudet, 1995; see http://www2.unil.ch/popgen/softwares/fstat.htm).

Population differentiation

Analyses of the partition of the genetic variance (AMOVA) were performed using ARLEQUIN v 3.5.1.3 (Excoffier, Laval & Schneider, 2005) from the 17 sampling sites. Pairwise divergences among samples were estimated using FST (Weir & Cockerham, 1984) implemented in the same software. Hierarchical partitioning was conducted in order to compare the molecular variation (1) among and within the 17 sampled locations and, (2) among and within the sampled hosts (wheat, barley, oat, and maize). A Bayesian clustering analysis was performed to determine the structuring of populations on the software STRUCTURE version 2.3 (Pritchard, Stephens & Donnelly, 2000) using the admixture ancestry and the correlated allele frequency models. The number of clusters (K) was set from 1 to 10 and repeated 20 times. Each repetition consisted of a burn-in period of 100,000 iterations and 1 million Markov Chain Monte Carlo iterations. The online program STRUCTURE HARVESTER (Earl & VonHoldt, 2012) was used to calculate the most probable number of genetic clusters (K) using the Evanno method (Evanno, Regnaut & Goudet, 2005). The graphical display of the genetic structure was produced using DISTRUCT (Rosenberg, 2004). The phylogenetic relationship between sampling localities was visualized using a neighbor-joining tree based on Cavalli-Sforza’s chord distance (Dc) between samples and plotted using FIGTREE version 1.3.1 (http://tree.bio.ed.ac.uk/software/figtree/). This tree was associated with the Bayesian clustering analysis in order to establish the phylogenetic relationship between the resulting genetic clusters.

Aphid reproductive performance

The MLG of R. padi found at the highest frequency in all samples (Rp1) was further studied in terms of its reproductive performance across different cereal hosts under aforementioned described laboratory rearing conditions (see “Aphid Collection and Laboratory Rearing” section). This would allow estimating any effect of the location and host of provenance. Thus, five clonal lineages of Rp1 were obtained: Rp1 genotype collected in Licantén from T. turgidum, Rp1 collected in Docamávida from barley, Rp1 collected in Limávida from T. turgidum, Rp1 collected in Villa Alegre from T. turgidum, and Rp1 collected in Cato from T. turgidum. These clonal lineages were maintained on barley for several generations before subjecting each clonal lineage to performance assessment. The reproductive performance was assessed on seedlings of oat (Avena sativa cv. Supernova), barley (H. vulgare cv. Sebastián), winter wheat (T. aestivum cv. Pantera) and durum wheat (T. turgidum subsp. durum cv. Llareta). When all seedlings attained growth stage 13 (three leaves unfolded; Zadoks, Chang & Konzak, 1974), each seedling was infested with five fully developed apterous adult aphids of the Rp1 genotype. Plants were placed in trays each one with six pots and distributed randomly in shelfs in the plant growth room. To avoids bias in the conditions, trays were randomly redistributed twice during the experiment. After 16 days, all surviving aphids were counted, separating apterous and alates individuals. The reproductive success was estimated by calculating daily per capita growth rate of aphids (PGR = dN/Ndt) as (ln(Nf) – ln(Ni))/((tf – ti)). Here, Ni and Nf indicate the initial and final number of individuals, respectively, while tf–ti correspond to the difference in days from the beginning to the end of the experiment (Kersch‐Becker & Thaler, 2019). Five replicates were included for each host. Number of nymphs, number of adult apterous and number of alates produce after 14 days were analyzed by two-way analysis of variance (factors: clonal lineage and hosts) using generalized linear models with according error distributions and link functions: Poisson distribution for count data (nymphs, apterous, and alates) and inverse gaussian distribution for continuous data (PGR). Generalized linear models were conducted using the interface Rcmdr implemented in the R statistical package 3.3.0 (R Core Team, 2012). The lsmeans package was used to conduct Tukey’s HSD multiple comparisons test also implemented in R package.

Results

Clonal and genetic diversities

A total of 377 R. padi individuals were genotyped at six microsatellite loci. All six microsatellite loci were reproducible and polymorphic in all samples, finding a total of 69 alleles (full data set available in Supplemental Information). The mean number of alleles at each sample site ranged between 1.7 and 5.2 (Table 2), with an average of 11.7 alleles per locus, and all were found in HW equilibrium. Evidence for null alleles was found for locus R5.138 (the allele frequencies ranged between 0.001 and 0.805). Regarding the genetic diversity, the observed heterozygosity (HO) ranged between 0.400 and 0.667, whereas the expected heterozygosity (HE) was between 0.354 and 0.684. Both indices of genetic diversity were homogeneous among locations.

Table 2 Population genetic parameters in populations of Rhopalosiphum padi.

Locality	N	G	G/N	D*	ED*	Na	HO	HE	FIS	
MP	30	8	0.267	0.634	0.444	2.3	0.604	0.452	−0.374	
ME	21	9	0.429	0.833	0.672	3.3	0.500	0.484	0.037	
SV	25	9	0.360	0.817	0.712	4.2	0.481	0.614	0.248	
LR	25	10	0.400	0.807	0.612	5.0	0.467	0.637	0.326	
LC	21	2	0.095	0.095	0.000	1.8	0.667	0.354	−0.900	
SC	30	8	0.267	0.703	0.586	5.2	0.583	0.684	0.176	
RA	12	5	0.417	0.727	0.510	4.0	0.400	0.630	0.405	
LI	53	14	0.264	0.818	0.751	3.7	0.524	0.548	0.042	
LIM	4	1	0.250	0.000	−1.00	1.7	0.667	0.333	−1.000	
DO	4	1	0.250	0.000	−1.00	1.7	0.667	0.333	−1.000	
CU	10	6	0.600	0.778	0.000	2.5	0.583	0.428	−0.238	
VP	67	14	0.209	0.704	0.596	4.7	0.583	0.585	0.044	
PE	46	13	0.283	0.747	0.595	3.8	0.564	0.560	0.096	
SnC	16	7	0.438	0.792	0.576	2.7	0.643	0.468	−0.434	
QE	7	3	0.429	0.667	0.563	2.3	0.667	0.481	−0.477	
VA	3	1	0.333	0.000	−1.00	1.7	0.667	0.333	−1.000	
CA	3	1	0.333	0.000	−1.00	1.7	0.667	0.333	−1.000	
Total	377	62	0.165	0.744	0.648	3.04	0.584	0.486	−0.137	
Note:

Localities name, number of aphids analyzed (N), number of multilocus genotypes (G), clonal diversity index (G/N), clonal heterogeneity indices (D*, adapted Simpson’s index), clonal evenness index (ED*, Simpson’s evenness index). Mean number of alleles (Na), heterozygosity expected (HE), heterozygosity observed (HO), and inbreeding coefficient (FIS) over all loci.

The combination of all six microsatellites allowed the identification of 62 MLGs. Among them, 23 MLGs were found more than once in the whole sample and were considered as multicopy, while 39 MLGs were unique (Table 2; Fig. 1). The number of MLGs in each location ranged from 1 to 14. Samples from LI, VP, and PE contained the higher number of MLGs (14, 14, and 13, respectively). Differently, samples from LIM, DO, VA, and CA localities only presented one MLG (Table 2).

Figure 1 Sampling sites of R. padi in central Chile.

The names of the localities are: MP, María Pinto; ME, Melipilla; SV, San Vicente; LR, Lago Rapel; LC, Las Cabras; SC, Santa Cruz; RA, Rauco; LI, Licantén; LIM, Limavida; DO, Docamavida; CU, Curepto; VP, Villa Prat; PE, Pencahue; SnC, San Clemente; QE, Quepu; VA, Villa Alegre; CA, Cato. Charts with many pie segments of colors show percentages of each multilocus genotype, and a gray area shows percentages unique multilocus of the samples of R. padi.

Most of the multicopy MLGs showed significantly low Psex values, suggesting very low chances of arising from sexual reproduction (Table 3). Mean clonal richness reached 0.162 (Table 2), while mean clonal diversity was 0.165 (Table 3), while. The clonal diversity according to the sampled locality ranged between 0.095 and 0.600, albeit similar among most locations (Table 3). Besides, the gross clonal diversity among hosts was also low (0.429, 0.250, 0.417, and 0.164 on oat, barley, maize, and wheat, respectively; data not show). LD between pair of loci was found in six out 15 cases (Table S2). Geographic distribution of the three most common MLGs revealed that were simultaneously found in only five localities, while the most common MLG (Rp1) was found in all sampled localities.

Table 3 Genetic features of repeated multilocus genotypes of Rhopalosiphum padi populations in central Chile.

MLG	N	%	P-value of Psex	R550	R5138	R3171	R273	R510	R529b	
Rp1	185	49.1	<0.001	322/322	251/251	231/241	283/301	273/275	187/193	
Rp2	29	7.7	<0.001	322/322	249/249	231/241	283/301	273/275	187/193	
Rp3	27	7.2	<0.001	322/322	251/251	231/241	283/301	269/275	187/193	
Rp4	20	5.3	<0.001	322/322	251/251	231/231	283/301	273/275	187/193	
Rp5	10	2.7	<0.001	322/322	249/249	231/231	283/301	273/275	187/193	
Rp6	9	2.4	0.058 ns	322/322	251/251	231/241	283/301	273/273	187/193	
Rp7	8	2.1	<0.001		251/251	231/241	283/301	273/275	187/193	
Rp8	6	1.6	<0.001	310/322	219/251	249/257	285/289	271/271	199/199	
Rp9	5	1.3	<0.01	322/322	251/251	231/231	283/301	269/275	187/193	
Rp10	5	1.3	0.515 ns	322/322	251/251	231/241	283/301	275/275	187/193	
Rp11	4	1.1	<0.001	322/322	251/251	231/241	283/301		187/193	
Rp12	3	0.8	<0.001	310/322	219/219	249/257	285/289	271/271	199/199	
Rp13	3	0.8	<0.001	322/322	249/249	231/241	283/301		187/193	
Rp14	3	0.8	<0.001	322/322	251/251	229/229	309/309	263/263	179/179	
Rp15	3	0.8	<0.001	322/322	251/251	231/231		273/275	187/193	
Rp16	3	0.8	<0.001	324/324	251/251	231/241	283/301	273/275	187/193	
Rp17	3	0.8	<0.001		251/251	231/241	283/301	269/275	187/193	
Rp18	2	0.5	<0.01	322/322	249/249	231/231	283/283	273/275	187/193	
Rp19	2	0.5	<0.01	322/322	249/249	231/231	283/301	269/275	187/193	
Rp20	2	0.5	0.072 ns	322/322	249/249	231/241	283/283	273/275	187/193	
Rp21	2	0.5	<0.05	322/322	249/249	231/241	283/301	269/275	187/193	
Rp22	2	0.5	<0.001	332/344	225/225	253/253	279/283	275/275	181/181	
Rp23	2	0.5	<0.001	334/334	251/251	231/241	283/301	273/275	187/193	
Unique	39	10.3								
Note:

PSEX is the probability that replicates of the same MLG are products of different sexual reproductive events.

Genetic differentiation among populations

Considering all copies for each MLG, the AMOVA showed significant genetic differentiation among all localities (FST = 0.0277; P = 0.001). Considering one single copy per MLG, however, the AMOVA resulted in low non-significant genetic differentiation among all localities (FST = 0.00931; P = 0.086). Because using multiple copies generates that distortions, the latter AMOVA was preferred. Consistent with this result, most of pairwise FST comparisons exhibited non-significant genetic differentiation among samples from different localities (Fig. S1). However, samples from RA, which included aphids collected from maize, showed higher pairwise FST, although most of these values were non-significant. The AMOVA considering samples from different hosts showed a significant genetic differentiation between hosts oat, barley, maize, and wheat (FST = 0.062; P = 0.034). The pairwise FST between samples from maize and wheat were the only statistically significant (FST = 0.160; P < 0.001).

The Bayesian analysis agreed with the AMOVA conducted with unique MLGs, showing there is no genetic differentiation among them (FST = 0.018; P = 0.077). Analysis of the population genetic structure considering one copy per MLG from all localities, revealed the best partition of the dataset involves two genetic clusters (K = 2) according to the Evanno method (modal value of ΔK, Fig. S2). Cluster 1 included individuals present in all localities, although six localities represented over 94% of this cluster. Additionally, Cluster 2 included individuals mainly from the central zone of study and maize samples (RA). This cluster also included other seven localities (SC, LR, SV, PE, LI, ME, and VP), without any pattern of geographic distribution. The phylogenetic relationship between samples from different localities revealed no grouping according to the geographical origin of samples (Fig. 2). Samples grouped more consistently in relation the clusters resulting from the Bayesian analysis. Samples from RA showed again a distinctive set of individuals exhibiting a high membership coefficient to cluster 2.

Figure 2 Neighbor-Joining tree and Bayesian clustering of R. padi based on sampled sites.

(A) NJ tree based on Cavalli-Sforza’s chord distance (Dc) of 17 sample sites of R. padi from various cereals crops in central Chile. (B) Bayesian clustering analysis of different sampling sites using STRUCTURE (version 2.3.2) software based on six microsatellite loci (sites organized from north to south distribution). The vertical lines are broken into colored segments showing the proportion of each individual assigned to each of the inferred K (K = 2). Geographic regions from which the populations belong appear along from north to south (1, MP; 2, ME; 3, SV; 4, LR; 5, LC; 6, SC; 7, RA; 8, LI; 9, LIM; 10, DO; 11, CU; 12, VP; 13, PE; 14, SnC; 15, QE; 16, VA; 17, CA). Charts with two pie segments show the results of Bayesian clustering analysis using STRUCTURE, percentages of cluster 1 (brown area) and percentage of cluster 2 (yellow area).

Aphid reproductive performance

Reproductive performance was characterized in the most frequent genotype Rp1 collected from four different hosts. Notice that performance was also assessed separately on Rp1 individuals generated from colonies collected from different sites but similar host (wheat). Number of nymphs produced by the most frequent genotype varied among hosts (main host effect: F3, 92 = 568.3; P < 0.001), with significantly higher values on T. turgidum and lower values on T. aestivum (Fig. 3A; Table S3). Nymphs also varied among clonal lineages of Rp1 (main clonal lineage effect: F4, 95 = 150.1; P < 0.001), with significantly higher values exhibited by Rp1-Cato and Rp1-Licantén and the lowest values exhibited by Rp-1-Limávida (Fig. 3A; Table S3). There was a significant clonal lineages × host interaction (F12, 80 = 24.4; P < 0.001), which was produced due to a lower number of nymphs produced by Rp-1-Villa Alegre lineage on T. turgidum (Fig. 3A; Table S3). Number of adult apterous and alates followed a similar trend, albeit values were about ten times lower than nymphs produced (Table S3). The PGR, which includes nymphs, adult apterous, and alates, also varied among hosts (main host effect: F3, 96 = 11.27; P < 0.001), with significantly higher values on T. turgidum and lower values on T. aestivum (Fig. 3B; Table S3). PGR varied among the five clonal lineages (F4, 92 = 9.38; P = 0.001), although no interaction between factors was found (F12, 80 = 8.45; P = 0.490).

Figure 3 Performance of the Rp1 genotype of R. padi.

(A) Number of nymphs (mean ± SE) and (B) population growth rate (PGR) (mean ± SE) after 14 days of infestation on four cereal hosts by five clonal lineages of the Rp1 genotype. Different letters within a host represent statistically significant differences according to Tukey’s HSD test (P < 0.05).

Discussion

Genetic features of R. padi in Chile suggest a primary reproductive mode of obligate parthenogenesis

We found that Chilean populations of R. padi exhibited low clonal diversity and low genetic diversity compared with sexual populations from elsewhere around the world. These findings are typical of populations reproducing predominantly by obligate parthenogenesis (see Table S1 in Figueroa et al., 2018). There are several factors that may explain the lack of sexual reproduction of R. padi in populations distributed in south-central Chile: (1) specific climatic conditions, which are characterized by mild winters, (2) the absence or low abundance of the primary host in the areas where cereals are grown, (3) spontaneous sex loss due to mutations or hereditary distortions (Frantz et al., 2005), (4) interspecific hybridization events (Lynch, 1984; Sunnucks et al., 1996; Sunnucks & Hales, 1996) possibly between R. padi and R. maidis, and (5) gene flow of asexuality genes from asexual populations to sexual populations. Reproduction between sexual and androcyclic individuals has been reported for R. padi (Delmotte et al., 2001, 2002, 2003; Halkett et al., 2008). All of these factors may be operating on Chilean populations of R. padi.

The first reports of R. padi lineages present in Chile were made in the 1960s (Zuñiga, 1986). One may speculate that the first-arriving individuals were subjected to a founder effect and probably lost the sexual phase of their original life cycle and hence the chance to alternate clonal and sexual reproductive phases. Alternatively, the lineages that arrived in Chile may have been those within a native range that were widespread and were reproducing most frequently via asexual reproduction (Figueroa et al., 2018). Our data show that the genotypic differences among Chilean MLGs of R. padi are very small and likely arose from mutations within a clonal family. For example, genotypes Rp1 and Rp2 differ simply in alleles of locus R5138 (251/251 and 249/249, respectively), which suggests the occurrence of a single mutational step. Similarly, the differences between Rp1 and Rp3 are small and again restricted to only a single locus R510 (273/275 and 269/275, respectively), which, in this case, suggests two mutation steps (269 -> 271 -> 273 -> 275). These events probably took place in Chilean populations of the grain aphid Sitobion avenae and the pea aphid Acyrthosiphon pisum (Figueroa et al., 2005; Peccoud et al., 2008), for which widely distributed genotypes are also frequently present in their native range. In addition to the recent introduction of genotypes, one must consider that some mutations in asexual aphids can yield genetic inheritance variations that, in turn, can give rise to a potentially selectable phenotypic variation (Wilson, Sunnucks & Hales, 2003). However, additional studies are necessary to assess the origin of R. padi genotypes and whether diversification via mutations has taken place in the Chilean populations of R. padi.

Worldwide variations in the reproductive mode of R. padi

The variation in reproduction modes exhibited by R. padi is a good example of how aphid populations maintain both sexual/asexual reproduction in their range of origin and how asexuality prevails in areas where these insects have been introduced. For example, European populations of R. padi are composed of sexual/asexual clones. In France, the genetic diversity, population structure, and transitions between reproductive modes have been extensively studied (Simon et al., 1996; Delmotte et al., 2001). Specifically, roughly 54% of asexual populations were copies of genotypes in the northern half of France, with seven of them widely distributed and persistent through time (Delmotte et al., 2002). Likewise, Halkett et al. (2005) studied the western population of R. padi in France and found the presence of two genetically distinct clusters composed of sexual and facultative asexual lineages. The latter consisted of few genotypes with numerous copies. More recently, an east–west transect in northern France revealed that the most common genotypes (“superclones”) of R. padi were distributed in clines along a climatic gradient (Gilabert et al., 2015). In Germany, three primary genetic clusters of R. padi were detected: “early colonizers,” found during winter, which largely disappeared later in the year; “late colonizers,” found on wheat fields and bird cherry trees and spreading mainly later in the year; and populations found exclusively on bird cherry trees (Klueken et al., 2012). Heteroecious, holocyclic populations of R. padi have been reported from northwest Russia, including clones capable of prolonged anholocyclic development (Vereshchagina & Gandrabur, 2016). In the UK, Leybourne et al. (2018) genotyped a very small sample of R. padi (n = 16) collected in Scotland and found a predominant genotype (genotype E) on three different cereal hosts. These results suggest that asexual reproduction persists as the primary mode of reproduction. In Spain, where the winters are not very cold and the primary host is absent, R. padi has been reported to overwinter parthenogenetically (Fereres et al., 1989; Pons, Comas & Albajares, 1993). In China, both reproductive forms of R. padi were also present; sexual populations were identified in spring wheat areas, and obligate parthenogenesis populations were found predominantly in winter wheat areas (Duan et al., 2015). A comparison between these populations revealed significant genetic differentiation; the cyclic parthenogenetic populations exhibited a larger number of alleles, greater allelic richness, and higher genotypic diversity compared with the asexual populations (Duan et al., 2017). Low levels of genetic diversity and differentiation using mitochondrial DNA were also detected in China (Wang et al., 2018b). Additionally, in the Southern Hemisphere R. padi displays only obligate parthenogenesis. The strong signature of obligates parthenogenesis in the Chilean populations of R. padi described in this work resembles that of the Australian populations of R. padi. Both populations presumably result from a recent invasion event. This event, together with mild winters and a low abundance of the primary host, may have resulted in the predominance of asexual reproduction (Figueroa et al., 2018; Valenzuela et al., 2010).

Given that aphid populations in similar environments tend to exhibit similar reproductive modes (Figueroa et al., 2018), it is very likely that other populations of R. padi in South America exhibit reproductive patterns similar to Chilean populations, particularly those from Argentina. We emphasize that more samples from the southernmost parts of Chile and Argentina are necessary. Interestingly, strictly parthenogenic populations of R. padi have been recorded on sub-Antarctic islands, which host several wild and introduced plants (Lebouvier et al., 2011; Delmotte et al., 2001). This finding strengthens the idea that asexuality is a successful aphid strategy for invading new habitats.

Weak host-based differentiation

We detected a weak degree of host differentiation among our samples (FST = 0.062). Specifically, only samples collected on maize in the locality RA exhibited a distinct set of genotypes (Fig. S1). A lack of host-based differentiation has been described in populations from Australia (Valenzuela et al., 2010) and France (Gilabert et al., 2009); these locales are where most genetic differences are associated with host alternation. Populations from China have been studied only from wheat samples, and a study using mitochondrial DNA furthermore confirmed low levels of genetic variation in this country (Duan et al., 2017).

Superclones of R. padi predominate in new areas of introduction

In addition to the obligate parthenogenesis of Chilean populations of R. padi described here, a large skewed frequency distribution of MLGs was found. One genotype (Rp1) dominated the sampling and was found in two consecutive seasons (2013–2014 and 2014–2015) in three different localities (Licantén, Villa Prat, and Pelarco). Genotypes Rp2 and Rp3 were also found consecutively in two seasons, but only in one location (Licantén). This situation has also been noted in other introduced aphid species such as Myzus persicae and S. avenae (Vorburger, Lancaster & Sunnucks, 2003; Loxdale et al., 2017). The presence of one dominant genotype in Australian populations of R. padi was even more striking; a single genotype accounted for 62.7% of the population (Valenzuela et al., 2010). Other studies performed in the Northern Hemisphere (e.g., France and China) have also demonstrated the predominance of superclones (Simon et al., 1996; Duan et al., 2017). However, R. padi persist only in areas with mild winters (e.g., northern France and central and northern China). Although R. padi is present in Chile in a wider range than even at more southern latitudes (Koch & Waterhouse, 2000), the area covered in our study was characterized by mild winters and encompassed where most cereals are produced (e.g., the Maule and Bio-Bio regions). Therefore, the presence of sexually reproducing R. padi aphids at more southern locations cannot be neglected.

The Rp1 clones studied exhibited variable performance across the cereals. Higher performance (i.e., a larger number of nymphs produced and PGR) on T. turgidum by Rp1 independent of provenance suggests some specific association with this host. T. turgidum has been found to be comparatively more resistant to R. padi than other cereal aphids because of its higher content of benzoxazinoid (Shavit et al., 2018). Interestingly, the Rp1 lineage sampled in locality Limávida exhibited the lowest reproductive performance on all hosts, with the exception on T. turgidum (Fig. 3). This finding suggests an intrinsic weakened performance independent of genetic background. It should be noted that aphid performance is dependent on the plant age and plant quality (Leather & Dixon, 1981; Stadler, Dixon & Kindlmann, 2002). Thus, at the plant stage used in this study, T. turgidum appears as the most suitable host for this R. padi genotype.

Several factors may underpin the phenotypic plasticity of the reproductive performance of aphids. Such plasticity appears to be particularly relevant in the case of introduced aphid populations (Figueroa et al., 2018; Loxdale, 2008). A recent screening of facultative endosymbionts of Chilean populations of S. avenae and R. padi revealed the presence, albeit at very low frequency of the bacteria Regiella insecticola (Zepeda-Paulo et al., 2018). However, it remains unclear whether that genotype corresponds with the Rp1 genotype described in this study. A recent investigation conducted in China (Guo et al., 2019) also screening facultative bacteria in R. padi and revealed the presence of seven species of facultative endosymbionts widely distributed over 32 R. padi populations. In that study, R. padi samples from Europe were compared with Chinese populations, and also described the presence of Hamiltonella defensa, Rickettsia sp., and Arsenophorus sp., which mostly exhibited multi-infections. In another recent study, a small population of R. padi from Scotland was shown to exhibit Hamiltonella defensa in two of the seven genotypes studied; the authors found that this endosymbiont conferred protection against the parasitoid Aphidius colemani in R. padi (Leybourne et al., 2018). Interestingly, this endosymbiont was present in the most common MLG genotype (Genotype E) and was the only genotype with a positive detection of three markers of APSE bacteriophages, which suggest that the presence of Hamiltonella defensa may confer protection against parasitoid wasps therefore ensure greater ecological success for this genotype.

The success of aphid superclones and challenges to pest management

Figueroa et al. (2018) reviewed the biological and genetic features of 23 different aphid species introduced in different geographic areas and climates. These authors reported that putative superclones were present in roughly 60% of species. The success of superclones in the introduced range, as in the case of the genotype Rp1 in Chile, may result from preadaptations in clonal lineages or neutral mutations that become favorable in the introduced environment (e.g., chemically defended hosts, managing practices). Because asexual lineages can rapidly accumulate mutations, obligate parthenogenetic genotypes can rapidly evolve closely related clonal lineages and persist in agroecosystems (Loxdale & Balog, 2018). Furthermore, superclones rapidly proliferate when they arrive in an agroecosystem. Nonetheless, agroecosystems are located in different biogeographic regions, they can exhibit highly similar conditions to the environment from where the aphids originated due to the homogenizing effects of agricultural practices.

The management of aphid pests exhibiting superclonality may be difficult, particularly in climate change scenarios (Figueroa et al., 2018). For instance, selection for and the spread of insecticide-resistant clones can result in very rapid changes in resistance levels in agricultural or horticultural systems (Figueroa et al., 2018; Simon & Peccoud, 2018). Therefore, aphids can quickly become a major problem when chemical and biological control fails owing to resistance. Resistance can be present even in aphids collected from a single host (Chen et al., 2013), which highlights the importance of selection on the rapid evolution of certain asexual lineages. Aphid superclones may possess insecticide resistance mechanisms, particularly in agroecosystems that receive frequent sprays of insecticides. For example, the superclone Burk1 of Aphis gossypii on cotton in west central Africa (Brévault et al., 2008, 2011) and superclones Aust-01 and Aust-02 on cotton in Australia (Chen et al., 2013) both carry the ACE1 mutation that confers resistance to pirimicarb and some organophosphate insecticides. Likewise, the widespread superclone NZ3 of M. persicae on potatoes in New Zealand (Van Toor et al., 2008) has enhanced carboxylesterase activity and kdr and super-kdr mutations that confer resistance to organophosphates and pyrethroids, respectively. Regarding R. padi, insecticide-resistant samples from China have been recently described (Wang et al., 2016, 2018a, 2018c). However, it remains unknown whether these strains correspond to clonal lineages developing as superclones.

The complex biology of aphid pests constitutes a challenge for crop protection. Several key features of aphid biology should be considered by farmers: (1) aphid pests can take advantage of the oversimplified design of current agroecosystems, (2) subjecting aphid populations to strong anthropic selection (e.g., insecticides, biological control) may result in the predominance of selected superclones that appear to be more aggressive, (3) factors such as life cycle strategies, environmental conditions (temperature, winter severity, regional variation), behavior (host specialization, flight behavior, and migration), and selection (massive use of insecticides and or natural enemies) affect aphid genetic variability, and (4) the evolution of that genetic diversity and population structure in time and space may be faster or slower depending on the intensity of the factors described above.

Conclusions

Chilean populations of R. padi are characterized by very low levels of genotypic and genetic diversity, suggesting that obligate parthenogenesis is the primary reproductive mode in the sampled range. Weak signs of genetic differentiation among localities and host-based differentiation were also observed. Among the MLGs found, one was present in all of the sampled populations at high frequencies and exhibited variations in reproductive performance on most common cereals cultivated in Chile. Chilean R. padi populations appear to be similar to those from Australia; both are composed of a single widely distributed superclone that likely resulted from a recent introduction. These results highlight the value of asexual reproduction during early stages of introduction to new regions in aphids, a distinctive feature that needs to be considered when implementing pest-management strategies.

Supplemental Information

Supplemental Information 1 Heatmap illustrating pairwise FST values among 17 samples of R.padi.

The range of colours from blue to white indicates decreasing pairwise genetic differentiation. Asterisks in bold are indicated the P-values that are below the significance value 0.000368, obtained after applying Bonferroni’s correction for multiple tests.

Click here for additional data file.

Supplemental Information 2 DeltaK values for different K calculated using the Evanno method.

Click here for additional data file.

Supplemental Information 3 Reproductive performance on different host of one R. padi clonal lineage.

Click here for additional data file.

Supplemental Information 4 Data of multilocus genotypes of the sampled R.padi individuals.

Allelle size for each locus, host plant collected, location and date of sampling.

Click here for additional data file.

Supplemental Information 5 Characteristics of the six microsatellite loci used to study of Rhopalosiphum padi:.

Locus name, central repeat motif, locus-specific hybridization temperature (Ta), and the number of alleles amplified with in each microsatellite locus.

Click here for additional data file.

Supplemental Information 6 Linkage disequilibrium analysis of Rhopalosiphum padi populations in central Chile.

P-value for each locus pair across all populations (Fisher’s method). Analyses were performed on dataset without repeated genotypes (one single copy per MLG). Asterisks indicates the P-values that are below the significance value 0.00033, obtained after applying Bonferroni’s correction for multiple tests.

Click here for additional data file.

Supplemental Information 7 Reproductive performance of five clonal lineages of the Rp1 genotype of Rhopalosiphum padi tested on four hosts.

Number of nymphs, apterous, alates and population growth of rate after 14 days of infestation under controlled conditions. Different letters within columns represent statistically significant differences according to Tukey’s HSD test (P < 0.05).

Click here for additional data file.

We thank Dr. Marco Cabrera-Brandt and Mr. Pablo Cordova for helping with the sampling, Mrs. Vickyana Navarro for her help with growing the samples under controlled-environmental conditions and conducting the performance experiments, and Mrs. Lucia Briones for her help with the molecular analyses.

Additional Information and Declarations

Competing Interests

Author Contributions

Data Availability

The authors declare that they have no competing interests.

María E. Rubio-Meléndez conceived and designed the experiments, performed the experiments, analyzed the data, prepared figures and/or tables, authored or reviewed drafts of the paper, approved the final draft.

Joceline Barrios-SanMartin conceived and designed the experiments, performed the experiments, authored or reviewed drafts of the paper, approved the final draft.

Felipe E. Pina-Castro performed the experiments, authored or reviewed drafts of the paper.

Christian C. Figueroa contributed reagents/materials/analysis tools, authored or reviewed drafts of the paper, approved the final draft.

Claudio C. Ramirez conceived and designed the experiments, performed the experiments, analyzed the data, contributed reagents/materials/analysis tools, prepared figures and/or tables, authored or reviewed drafts of the paper, approved the final draft.

The following information was supplied regarding data availability:

The raw data are available in the Supplemental Files.

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
