# Peer review of "Asexual reproduction of a few genotypes favored the invasion of the cereal aphid Rhopalosiphum padi in Chile"

_PeerJ, doi:10.7717/peerj.7366_

## Round 0.1 · original submission · Major Revisions

Dear Dr. Rubio-Meléndez and colleagues:

Thanks for submitting your manuscript to PeerJ. I have now received two independent reviews of your work, and as you will see, the reviewers raised several concerns about the research. Despite this, these reviewers are optimistic about your work and the potential impact it will have on ceareal aphis research. Thus, I encourage you to revise your manuscript, accordingly, taking into account all of the concerns raised by both reviewers.

Aside from the criticisms raised by the reviewers in their reports, be sure to enlist the assistance of an English expert before submitting your revision.

I look forward to seeing your revision, and thanks again for submitting your work to PeerJ.

Good luck with your revision,

-joe

[]

Reviewer 1 ·

Basic reporting

The article presents a nice comprehensive study of the genetic variation within Chilean R. padi populations, however there are some areas of concern which should be addressed

Notes to the author relating to their raw data:

Proportion of alate adult data: proportion data is bound at 0.0 and 1.00. Why does proportion data exceed 1.00, and go beyond 20? Is this percentage data?
Proportion of alate data: In host plant column should be . not , following the shortened genus names
Genotyping result data: data seems fine, but is hard to navigate/assess in current format I suggest the authors separate this into separate columns instead of a string of text in the first column.

Notes to the author relating to the introduction of the manuscript:
Line 77: "As described for several pest aphids, the predominance of a certain reproductive mode in R. paid relies on geographic factors". The authors should provide a citation for this statement and briefly describe the examples they mention, this will also improve the introduction by providing a wider assessment of the literature
Line 82 - 95: I recommend the authors put more emphasis on the importance of their study, and the reasoning behind why a) assessing the genetic diversity in R. padi populations in Chile is important; b) how this can help improve crop protection (or inform pest management strategies)
Line 92 - 95: I think the manuscript can be improved if the authors more clearly describe what they did in the study and provide a comprehensive description of the key findings. For example: "In this current study we examine the genetic diversity in populations of the cereal aphid, R. padi, in the main cereal growing region in Chile. We show that...."

Notes relating to the materials and methods:
Line 164 – 167: This information would fit better in the introduction
Line 236 – 237: For the PGR equation, please ensure that log (ln) is written as ln not Ln

Notes to the author relating to the discussion of the manuscript:
Section 355 - 402:
I suggest the authors also include the Leybourne et al 2018 (doi: https://doi.org/10.1111/1744-7917.12606) study in their assessment of European R. padi genetic diversity (this study assessed genotype distribution in small R. padi population in Eastern Scotland, UK)
Also, as a potential recommendation, the authors could improve the manuscript greatly by extracting the genotyping information from all the studies they mention and to examine whether their Rp1 genotype (most distributed in Chile) has a similar distribution in other countries.
Section 414 - 448:
Line 439 – 446: Guo et al 2019 (doi: https://doi.org/10.1002/mbo3.817) and Leybourne et al 2018 describe endosymbionts in R. padi, these are more suitable references than the Sepulveda et al 2017 reference as they show that R. padi is able to form these relationships. Additionally, the Zepeda-Paulo et al 2018 study shows that in one of their aphid clones assessed (named Rp1) a small proportion of the aphids contained the endosymbiont R. insecticola, so the statement that secondary symbionts have not been found in Chilean R. padi is incorrect.

Notes to the author relating to the Tables and Figures of the manuscript:
Table 1: Host plant – for wheat please indicate which were collected from durum wheat and which from aestivum wheat
Table 3: For microsatellite loci – suggest splitting the heterozygous numbers to show more clearly the size of each allele (so change from displaying 322322 to 322 | 322) this would facilitate ease of readability. The authors should also define Psex
Some figures are rather large – too large to print. Figure 1 and 2 exceed A3 size, so would not be able to be easily printed and observed
Fig. 1: Legend: typo – should read “unique MLG” not “uniques MLG”.
Fig. 2: It is unclear what the blue, green, red, grey, and gold segments of the graph are showing

Some suggested English language improvements throughout the manuscript
Line 43: Suggest deleting "in following years"
Line 46 - 49: Suggest the authors rephrase this wording of this section to make their point more clear
Line 53 - 55: Suggest rephrasing to "their reproduction mode, which is based on parthenogenesis and can result in rapid population increases"
Line 62: delete "aphids"
Line 64: delete "the aphid"
Line 68: delete "the aphid"
Line 82 - 84: This sentence needs to be rewritten to improve the readability, for example "Understanding the genetic features of pest populations can help to inform which pest management strategies would be most appropriate"
Line 99: Rephrase to “R. padi individuals were collected…”
Line 100 – 102: Rephrase to “Sampling was carried out during spring…”
Line 106 - 107: Rephrase to “Sampled plants include the most commonly cultivated cereal crops:…”
Line 116: Please use the correct entomological terminology of “apterous” when mentioning wingless aphids, and “alate” when referring to winged aphids
Line 117 – 120: Rephrase to: “In addition, live individuals were collected from each plant to establish clonal laboratory lines for aphid performance assessment”
Line 121: change “in barley” to “on barley”
Line 123: Define what a “transparent plastic mica” is, this is not a usual term before
From Line 123: expand this section to include information on how the collected aphid lines were reared in the laboratory (moving this information up from the aphid performance section)
Line 136: replace “in 2 min” with “of 2 min”
Line 201: Delete the “of” in the two instances of “within of”
Line 228: Please rephrase to “The reproductive performance of Rp1 was assessed on …”
Line 235: please use the terms apterous and alate
Line 307 – I think it is best to state “comparing its PGR when feeding on four different hosts” use of the term “reared on” implies that the aphids were kept on the four different plants in culture, not during the experiment.
Line 330: delete “exhibiting”
Line 331: Rephrase to “rather than act as casual agents”
Line 331: typo, should be “latter” not “later”
Line 330 – 338: I think in general this section needs to be rethought and rephrased, it isn’t explicitly clear how this point relates to the findings reported in the manuscript
Line 340: typo, should this not be “founder effect”?
Line 340 – 342: This appears to be speculation, the authors should indicate that they are speculating
Line 343: add “the” so that sentence reads “that in the native” delete “widely”
Line 343 – 344: rephrase to “were reproducing most frequently by asexual reproduction”
Line 345 – 346: The authors should include a reference to the four/three most widely distributed grain and pea aphid genotypes. It is not clear if this is supported by the Figueroa 2005 and Peccoud 2008 references, or if these references just support the French lineage statement.
Line 359: typo, should be “formed of”
Line 368: rephrase to “the most common genotypes”
Line 376: rephrase to “is not very cold”
Line 378: Rephrase to “However, in China”
Line 380: typo, should be “predominantly”
Line 386: Reword to “Additionally” not “Differently”
Line: 388: “Are similar to Australian populations”
Line 389 onwards: I suggest the authors make use of an English language editing service

Experimental design

Certain aspects of the experimental design need to be addressed (or clarified) to improve the article. Some detail which would be required for full reproduction of the work are currently missing from the manuscript. Key information regarding experimental design and experimental conditions is missing (specifically related to the genotype fitness experiment)

Line 92 - 95: The authors need to more clearly define the purpose of the research/the research question more clearly at the end of the introduction
From Line 218: include information on the host plant from which each Rp1 clonal line was originally collected from. This is mentioned in the results – line 309, I suggest stating this in the methodology section not the results section.
Line 221 – 228: I suggest moving the aphid rearing conditions to the previous section (indicated by my point above)

Specific notes relating to the materials and methods section of the manuscript
Refer to Table 1 (the 17 sampling sites) in the methodology section where appropriate to facilitate reading
Line 104 – 105: What was the selection criteria which determined which of the initial fields were sampled in the subsequent sampling campaigns? Why weren’t all the fields revisited in the other years?
Line 129 – 130: To facilitate full reproducibility of the experiment please provide the Forward and Reverse primer sequences used in this study (even if they are reported in the Simon et al 2001 study). This can be in the form of an additional supplementary table
Line 135: What concentration (active Units of enzyme) of Platinum Taq were used?
Line 136: How was DNA concentration determined? And were 50 ng of DNA used in total, or 50 ng / ul
Line 140 – 143: please provide the full methodology for the capillary electrophoresis procedure, for example: were the PCR products diluted? How much of the sample was loaded into the reaction solution? How much dye was used (and what type of dye was used). What size standard was used? Were the samples suspended in Hi-Di Formamide?
Line 218 – 220: Please provide detailed information on the conditions used for these experiments – light intensity, temperature etc. Were the assays conducted in glasshouse or controlled environments? Assays carried out on the lab bench will not be appropriate as these conditions aren’t controlled
Line 229 – 231: Please indicate at which developmental stage (using the Zadoks et al staging key: https://dx.doi.org/10.1111/j.1365-3180.1974.tb01084.x) all plants were when they were used in the fitness experiment. Plant developmental stage can effect aphid fitness (for e.g. Leather and Dixon 1981 https://doi.org/10.1111/j.1744-7348.1981.tb03006.x and Stadler et al 2002: https://doi.org/10.1046/j.1461-0248.2002.00300.x). The results are only comparable across plant species if the experimental plants were at the same developmental stages. The authors should also discuss these results in relation to the findings of the Leather and Dixon paper mentioned above.
Line 232: Please provide the growth and environmental conditions of these controlled environment chambers
Line 233 – 242: Please provide information on whether this was glasshouse, field, or controlled environment experiment. Please provide explanation how aphids were contained onto each plant to prevent their escape
Line 234 – 235: Was the age of the adult aphids synchronised before they were used in the fitness experiments?

Validity of the findings

Genotyping results and interpretation - in general, I believe that these aspects of the study have been carried out robustly, analysed appropriately, and interpreted correctly. Besides the points mentioned above regarding providing more information on the experimental design, I have no major criticisms relating to this aspect of the work.

Aphid reproductive performance - there are several aspects of this part of the manuscript which need to be addressed and clarified before it can be determined whether this aspect of the study was robust and statistically sound.

Notes on the statistical analysis of the aphid performance data
Replication level of five for aphid fitness data is rather low, however as this has to be weighed against the number of individual treatments (20) as such I believe this is adequate.

Line 233 – 242 – how were the reproductive experiments set out? Please include information on the experimental design of this (randomised, split-plot etc.). I suggest the authors re-run the analysis and take into account the experimental setup if possible, for example if the data is a fully randomised blocked design, I suggest the authors statistically account for the intra-block variation (for example, through the use of general linear/linear mixed effects models).
Line 239 – 242: How were the proportion data analysed? This isn’t mentioned in the manuscript. However, If the authors used two-way ANOVA (as they did with the PGR data) this isn’t the most appopiate statistical methodology to use for the analysis of proportion data. As proportion data is bound at 0.00 and 1.00 general linear models with binomial distribution (aka log-linear/logistic general linear models) are more appropriate for the analysis of data with this structure
Line 240: Was this Tukey HSD or Tukey LSD?
Line 241 – 242: What statistical software was used for this analysis?

Notes relating to the reporting of these results:
Line 310: the authors indicate that there are differences amongst the different host plants and on line 313 – 314 indicate that this is between Limavida lineage and all others (Fig. 4). It isn’t clear which lineages are significantly different from the others – The authors carried out Tukey HSD tests to test this, but do not report it. I suggest the authors either report the differences in this section in the text, for example by rephrasing the paragraph to state “The PGR varied among hosts (F3, 90 = 13.48; P < 0.001), with significant differences between Limavida and xxx (statistical results of HSD test)”. A simplified way of displaying this would either be in the form of an additional table reporting these which can be referenced to in the text, or by plotting the results of the post-hoc tests on Fig. 4.
Line 311: The same point as above should be taken on board by the authors with regards to the observed difference between the different aphid lineages. I suggest a table reporting the post-hoc results may be the most simple and appropriate way to report these results.
Line 311: “No interaction between factors was found” the authors should report the statistical results here, even if they are non-significant
Lines 315 – 317: Again, differences are indicated between the proportion of alate aphids in relation to host and lineage, but no post-hoc results are displayed to provide information on which differences these are. I suggest the inclusion of a "post-hoc" table detailing which factors are different for the PGR and the Alate proportion results.
Line 316: “but no interaction between both factors was found” the authors should report the statistical results here, even if they are non-significant
Fig. 4: This figure is not very clear, is hard to interpret and should be improved to aid the reader. Ideally, as they are not reported in the text, the post-hoc analysis needs to be included on the fig (or reported in a table if it is not going to be empirically stated in the manuscript). Fig. 4B is this percentage or proportion data? Proportion data would be bound at 0.00 and 1.00. Generally, the figure is hard to interpret. Other potential ways the authors could consider for displaying this data are:
o A simplified table stating the mean and error, this could also simply be edited to include the results of the post-hoc analysis.
o A dot plot – as sample number is low (5) dot plots will increase the transparency of the data. It would, however, be difficult to clearly display the post-hoc results on this graph
o Box plots – would make graph a lot larger, but would be greatly increase the readability of the figure and make it easier to interpret than current display, post-hoc results could also easily be added to this

Additional comments

In this study by Rubio-Meléndez et al. the genetic diversity of R. padi is assessed throughout Chile. Using a substantial sample size the authors show that one genotype, Rp1, is the the most prevalent in the regions assessed. The authors then carry out aphid fitness assays to examine the fitness of this aphid genotype in individuals collected from several regions.

The first half of the manuscript (analysis of aphid genetic diversity in Chile) has been adequately carried out and presents a nice, novel insight into the genetic diversity within Chilean R. padi populations. The prevalence of one genotype is specifically interesting. However, certain aspects of the aphid fitness experiment need rethinking and supplemented with addiitonal information as it is not currently possible to fully assess the validity of the findings as the experimental design of these experiments has not been adequately explained.

Overall, the manuscript needs a substantial rewrite to facilitate reading, use of English is confusing at times and sections of the discussion are often hard to understand. The overall quality of the manuscript would be improved by setting a research question/hypothesis in the introduction and using this as the basis of the discussion

Reviewer 2 ·

Basic reporting

The manuscript is well written and the paper is clear, but the text requires an additional round of polishing. Some typos I noticed include: sampling months in the Table 1 are written in Spanish, MICRO-CHEKER (Line 144), funder effect (Line 340), bluest (Line 790), interchangeable use of “its” and “their” is confusing (Line 92 – 95).

Using term “amplification” in the title may be misleading for the readers, as it might be related to DNA amplification. Also, term “clonal” is not the most suitable, as has been discussed by Loxdale (2008). Instead, “asexual” or “parthenogenic” could be used. The use of abbreviations (CP, OP, PGR) should be avoided, because they reduce readability of the text.

Please format the Main text and Supplementary Figures (consistent font style, font size, alignment, panel labels) to enhance representation.

Experimental design

Authors performed sufficient sampling for R. padi, the sampling scheme is adequate.
Number of replications for the analysis of reproductive performance is sufficient.

Readers would benefit from an explanation how 6 microsatellite loci were chosen and why their resolution is adequate (Line 130).

Authors assessed reproductive performance of the most common genotypes of R. padi on different host plants, but it is not explained (in the Introduction) which research question is addressed using this analysis.

While the Authors describe that the sampling happened during spring (Line 101), Table 1 indicates three sampling seasons: August 2013 – January 2014, September – November 2014, and August 2015. January 2014 does not belong to spring. The Authors write: “During the following years, the same set of fields were sampled” (Line 106) – but Table 1 suggests repeated sampling only for 3 localities (LI, VP, PE). The text in the Materials and Methods should be corrected to match the Table 1.

Methods described with sufficient detail, for the most part. Please provide details how the level of photosynthetically active radiation has been measured and controlled (Line 125).
Software used for the analysis of reproductive performance should be mentioned (Lines 240-242). Please also provide the raw data and code for this analysis to ensure reproducibility.

Validity of the findings

Repeated genotypes found in the study may result from cross-contamination between the samples. In the Materials & Methods, it is not mentioned how the authors controlled for the cross-contamination. Negative control (PCR master mix without DNA template) should have been run on a gel together with other samples to rule out potential cross-contamination. Please provide information how cross-contamination was controlled.

Are the same genotypes over-represented in different sampling seasons? This analysis is feasible for at least first two seasons: August 2013 – January 2014 and September – November 2014. This will help to assess the role of obligate parthenogenesis in explaining the finding of repeated genotypes.

For the linkage disequilibrium analysis (Line 180), correction for multiple testing should be applied. As highlighted by Halkett et al (2005) the LD analysis should be performed on datasets without repeated genotypes. It is unclear (Line 180) if the results shown in Table S1 were performs on dataset with or without repeated genotypes. Please provide details.

For the pairwise analysis of Fst (Line 198-202), correction for multiple testing should be applied. Please explain which formula has been used for FST estimation – and why it produces unusually low values (Figure S1).

Previously it has been reported (Delmotte et al 2002) that asexual lineages of R. padi form a distinct phylogenetic cluster. However, Figure 2 and Figure 3 suggest that candidate asexual lineages are polyphyletic. This contradiction should be addressed.

---

## Round 0.2 · Minor Revisions

Dear Dr. Rubio-Meléndez and colleagues:

Thanks for revising your manuscript. The reviewers are very satisfied with your revision (as am I). Great! However, there are several remaining concerns by both reviewers. Please address these ASAP so we may move towards acceptance of your work.

Best,

-joe

Reviewer 1 ·

Basic reporting

Following revision the manuscript is more clear, the addition of the paragraph at the end of the introduction (from line 103) greatly increases the readability.

I have made some additional (minor) suggestions below:

Line 118 - 119, consider rephrasing for clarity
Line 201: indices not indexes
Line 260: Consider rephrasing from "plants were disposed" disposed implies they were thrown out
Line 500: H. defensa not H. insecticola

Tables: For Tables reporting statistical results, I recommend the authors use <0.001 instead of 0.0000

Supplementary Data 1: Column C should read "nymphs"

Experimental design

No comment

Validity of the findings

No comment

Additional comments

In general, I believe the authors have made ample changes to the manuscript and have taken into account the comments and concerns raised by myself and the other reviewer.

Reviewer 2 ·

Basic reporting

The text of the manuscript has been improved, couple more typos could be corrected:

Obligated parthenogenesis -> obligate parthenogenesis? (Line 106 of the PDF)
Partenogenesis -> partenogenetic? (Line 95 of the PDF)
SupplementaryData1-Performance.csv: Nymfas -> Nymphs?
Figure 1, Legend: Uniques MLG -> Unique MLG?

The figures have been improved as well. I have further suggestions for improvement of the Figure 2A.

According to the Table 2, and the Figure 1, in the localities LIM, DO, VA, CA only the most frequent genotype Rp1 has been found. This agrees with STRUCTURE clustering: all the individuals from these localities belong to the cluster 1 (marked by the brown colour, Figure 2B). But in the Figure 2A, in the circles representing these localities there are tiny yellow sectors. I assume they denote the frequency of 0 of the genotypes of the cluster 2? Then these tiny yellow sectors should be removed, as they are misleading (i. e. they suggest both clusters are represented in the localities).

Figure 2B would greatly benefit from increased resolution. The rectangles which denote the localities should have thicker black boarders, to improve readability of the figure (currently it is hard to distinguish between the localities). Please also use arrows to label localities with small number of individuals (LIM, DO, VA, CA).

Experimental design

Please mention and cite the specific R package used for the analysis of reproductive performance (Line 273-274 of the PDF).

Validity of the findings

Results:
Please report the frequencies of the most common genotypes in different seasons (454-457 of the PDF).

Table S2:
The reported significance level of 0.00033 has an extra 0, it seems to be a typo (for the Bonferroni correction 0.05/15 = 0.0033). Also, please explain why the p-values slightly changed in the revised version of the manuscript?

Eliminated Figure 3:
I suggest it is worthwhile to keep this figure at least as a supplementary material as it gives a good overview of how repeated and unique genotypes are related. I assume that the choice of colors used to mark “clades” is arbitrary (i.e. it does not relate to the Figure 1), so they could be eliminated. If my assumprion is wrong, please provide explanaton what do these colours denote.

Additional comments

The Authors provided a detailed response to all the comments and improved the manuscript.
Their new result of finding that the same most common genotypes were present in different seasons strengthens their conclusions about asexual reproduction of R. padi in Chile.

---

## Round 0.3 · accepted · Accept

Dear Dr. Rubio-Meléndez and colleagues:

Thanks for revising your manuscript to PeerJ, and for addressing the concerns raised by the reviewers. I now believe that your manuscript is suitable for publication. Congratulations! I look forward to seeing this work in print, and I anticipate it being an important resource for research communities studying cereal aphid biology and control.

Thanks again for choosing PeerJ to publish such important work.

-joe